

# The updated Multi-Model Large Ensemble Archive and the Climate Variability Diagnostics Package: New tools for the study of climate variability and change

Nicola Maher[1,2], Adam S. Phillips[3], Clara Deser[3], Robert C. Jnglin Wills[4], Flavio Lehner[3,5,6], John Fasullo[3], Julie M. Caron[3], Lukas Brunner[7], and Urs Beyerle[4]

[1]Australian National University, Canberra, Australia
[2]Cooperative Institute for Research in Environmental Sciences, University of Colorado at Boulder, USA
[3]Climate and Global Dynamics Laboratory, National Center for Atmospheric Research, Boulder, CO, United States
[4]Institute for Atmospheric and Climate Science, ETH Zurich, Zurich, Switzerland
[5]Department of Earth and Atmospheric Sciences, Cornell University, Ithaca, NY, United States
[6]Polar Bears International, Bozeman, MT, United States
[7]Affiliation: Research Unit Sustainability and Climate Risk, Center for Earth System Research and Sustainability (CEN), University of Hamburg, Hamburg, Germany

**Correspondence:** Nicola Maher (nicola.maher@anu.edu.au)

**Abstract.**

Observations can be considered as one realisation of the climate system that we live in. To provide a fair comparison of climate models with observations, one must use multiple realisations or *ensemble members* from a single model and assess where the observations sit within the ensemble spread. Single model initial-condition large ensembles (LEs) are valuable tools

for such an evaluation. Here, we present the new multi-model large ensemble archive (MMLEAv2) which has been extended to include 18 models and 15 two-dimensional variables. Data in this archive has been remapped to a common 2.5 x 2.5 degree grid for ease of inter-model comparison. We additionally introduce the newly updated Climate Variability Diagnostics Package version 6 (CVDPv6), which is designed specifically for use with LEs. The CVDPv6 computes and displays the major modes of climate variability as well as long-term trends and climatologies in models and observations based on a variety of fields. This

tool creates plots of both individual ensemble members, and the ensemble mean of each LE including observational rank plots, pattern correlations and root mean square difference metrics displayed in both graphical and statistical output that is saved to a data repository. By applying the CVDPv6 to the MMLEAv2 we highlight its use for model evaluation against observations and for model inter-comparisons. We demonstrate that for highly variable metrics a model might evaluate poorly or favourably compared to the single realisation the observations represent, depending on the chosen ensemble member. This behaviour

emphasises that LEs provide a much fairer model evaluation than a single ensemble member, ensemble mean, or multi-model mean.



## 1 Introduction

Single model initial-condition large ensembles (LEs) are a powerful tool for understanding past, current and future climate (e.g. Deser et al., 2020; Maher et al., 2021a). A single LE allows for both the quantification and separation of the forced
response (response of any given variable to external forcing) and unforced internal variability, while the availability of multiple LEs additionally enables the assessment of model differences in both quantities (Deser et al., 2020; Lee et al., 2021; Maher et al., 2021b; Wood et al., 2021; Maher et al., 2023). LEs also facilitate a robust evaluation of individual climate models in comparison to observations, as they include a range of possible climate realisations (Goldenson et al., 2021; Suarez-Gutierrez et al., 2021; Labe and Barnes, 2022). Observations can be considered as one realisation of the climate system, and as such
the interpretation of its comparison to a single historical simulation from a climate model, an ensemble mean, or multi-model mean is complicated. To this point, fairly evaluating projections in climate models, or highly variable climate quantities against this single realisation of the real world is only possible by either taking long time averages, to effectively smooth out natural climate variability and allowing for the assessment of the model's forced response, or by using a LE and evaluating whether observations sit within the model's ensemble spread. This makes LEs invaluable tools for model evaluation. The value of LEs
derives from their large sample size. In addition to providing a range of plausible outcomes arising from the superposition of forced response and internal variability, the value of LEs derives from the sheer volume of data that they provide, enabling robust statistics on climate variability. For example, 250 years of simulation (e.g. 1850-2100) in a LE of only 20 members yields 5000 years of data for analysis, while a typical pre-industrial control simulation (piControl) provides 500-2000 years of output.

The Multi-Model Large Ensemble Archive (MMLEA; Deser et al., 2020) was the first compilation of many LEs. The overview paper has been cited over 650 times despite only being published in 2020 (4 years ago; data as of 22/10/24). This highlights the community's need for and the value of such an archive. In this paper, we present the MMLEAv2 archive that has been expanded beyond the original MMLEA to both include more models (18 compared to the original 7, largely those from the more recent CMIP6 archive), and 7 additional two dimensional variables. The MMLEAv2 and a suite of observational data sets
have also been regridded onto a 2.5 degree common horizontal grid to reduce data size, and to allow for straightforward model-to-model, and model-to-observations comparison. The MMLEAv2 allows for straightforward comparison between multiple models, and observations, with scope for scientists to download additional data on the native grids from the original data sources to supplement analysis.

The MMLEAv2 is an ensemble of opportunity, including both CMIP5 and CMIP6 class data, with no consistent forcing
scenario available for all models. While this provides a limitation for the dataset, possibilities exist to compare warming levels rather than comparing across time (Seneviratne et al., 2021). This enables a more direct comparison between different scenarios, and circumvents this limitation, although it can only be implemented where the warming level itself, not the warming trajectory is important (Hausfather et al., 2022).

An effective way to explore the characteristics of internal variability and forced response in the MMLEAv2 archive is to use
the National Science Foundation National Center for Atmospheric Research Climate Variability Diagnostics Package Version6





(CVDPv6; Phillips et al., 2020). Previous versions of this package have been used to investigate how modes of variability are represented over generations of CMIP climate models (Fasullo et al., 2020), questions of model evaluation robustness given the limited duration and certainty in observational datasets (Fasullo et al., 2024), and strengths and weaknesses of US climate models in simulating a broad range of modes of variability (Orbe et al., 2020). In this study, we present the latest version of this package and highlight its value specifically for use with LEs, although it can also be applied to control simulations and models with one realisation. This automated package allows for model comparison with multiple observational data sets, as well as inter-model comparisons, with diagnostics completed over time periods of the user's choosing. The CVDPv6 provides for an ensemble mean view in addition to individual member analysis. The ensemble mean view includes diagnostics of the forced component of climate variability and change, as well as metrics of the rank of the observations within the model ensemble spread to illustrate model bias. The package computes the leading modes of variability in the atmosphere and coupled ocean-atmosphere system including the El Niño Southern Oscillation (ENSO), as well as long-term trends, climatologies, and a variety of climate indices. All results are saved in both graphical and numerical form, allowing for subsequent analysis and display. The user can specify any set of model LEs and observational data sets (including multiple data sets for a given variable) over multiple time periods for analysis. Additionally, the CVDPv6 offers a range of detrending methods that can be applied before computing diagnostics. The versatility of the CVDPv6 makes it an efficient and powerful tool for analysing LE output. In addition, the CVDPv6 displays information from all model simulations and observational data sets on a single page, facilitating model inter-comparisons and assessment of observational uncertainty. In this paper, we will demonstrate the utility of the CVDPv6 as applied to the new MMLEAv2 and highlight some examples of its use and the insights derived therefrom.

The aims of this paper are threefold:

1. Introduce the MMLEAv2 and describe the data available in this extended archive;

2. Demonstrate the utility of the CVDPv6 and the insights one can derive from it;

3. Highlight the importance of using LEs for model evaluation.

## 2 The MMLEAv2 Dataset

The MMLEAv2 includes 18 models and 15 monthly variables as outlined in Table 1. The original MMLEA consisted of 7 CMIP5-class models with consistent historical and RCP8.5 forcing and 2 models with additional forcing scenarios. The MMLEAv2 contains 6 of the same models from the MMLEA as well as 12 additional CMIP6-class models. While a consistent RCP8.5 future forcing scenario was available for the CMIP5-class models, the CMIP6-class models use one or both of the historical plus SSP370 or SSP585 scenarios. Given that this is an ensemble of opportunity, whichever scenario was available and had the largest number of members is included in the MMLEAv2 (see Table 1). CMIP6-class model data is available for the period 1850-2100, except for GFDL-SPEAR-MED which covers the time period 1921-2100. CMIP5-class data is available for a range of time periods, with the longest simulation from 1850-2099 (MPI-GE) and the shortest from 1950-2100 (CanESM2). The LEs have a minimum of 14 ensemble members for the historical period and a maximum of 100 members. In addition to





the original variables in the MMLEA [precipitation (pr), sea level pressure (psl), surface air temperature (tas), horizontal wind stress in the u direction (tauu), horizontal wind stress in the v direction (tauv), sea surface height (zos), 500mb geopotential height (zg500)] the MMLEAv2 provides sea surface salinity (sos), evaporation (evspsbl), mixed layer depth (mlotst), sea-ice concentration (siconc/sic), and 3 monthly extreme indices computed from daily data. We note that unlike the MMLEA, the MMLEAv2 does not provide surface radiative variables. The 3 monthly extreme indices (Zhang et al., 2011) are the monthly maximum of the daily maximum temperature (TXx), the monthly minimum of the daily minimum temperature (TNn), and the monthly maximum of the daily precipitation (Rx1day); they are computed from daily tasmax, tasmin, and pr data using the monmax and monmin functions in Climate Data Operators (CDO; Schulzweida).

Data was compiled from the sources listed below and remapped to a common 2.5 x 2.5 degree grid. Conservative mapping was used for pr, psl, tas, tauu, tauv, TXx, TNn, Rx1day and zg500, whereas distance-weighted mapping was used for all other variables and for all variables in the CESM2 model due to an inability to use the other gridding tools on CESM2's native grid and PSL in GFDL-CM3 due to an additional grididng issue. We also provide remapped observational datasets on the same common grid for ease of use as outlined in Table 2. We note that the data in the CESM models was shifted a month earlier to resolve the issue of netcdf readers reading the data 1 month off, which is an issue on the CESM native temporal grid.

Data sources:

- ETH cmip6-ng archive (already regridded to a 2.5 degree grid by Brunner et al., 2020); variables pr, psl, tas, tauu, tauv, zos, siconc and TXx, TNn, and Rx1day for models ACCESS-ESM1-5, CanESM5, EC-Earth3, IPSL-CM6A-LR, MIROC6, MIROC-ES2L and UKESM1-0-LL and zg500 for MIROC-ES2L and UKESM1-0-LL

- The CMIP6 archive (https://esgf-node.llnl.gov/search/cmip6/); variables evspsbl, mlotst, sos for models ACCESS-ESM1-5, CanESM5, EC-Earth3, IPSL-CM6A-LR, MIROC6, MIROC-ES2L and zg500 for all these models except MIROC-ES2L

- The MMLEA (https://www.cesm.ucar.edu/community-projects/mmlea); CanESM2, CESM1, CSIRO-Mk3-6-0, GFDL-CM3, MPI-GE (pr, psl, tas, sic and TXx, TNn, and Rx1day)

- The CESM2 Large Ensemble (https://www.cesm.ucar.edu/community-projects/lens2); CESM2)

- The MPI-GE Archive (https://esgf-data.dkrz.de/projects/mpi-ge/); MPI-GE (tos, zos, tauu, tauv, z500, sos, evspsbl)

- Directly from Thomas Frölicher at the University of Bern; GFDL-ESM2M

- Directly from Dirk Olonscheck at the Max Planck Institute for Meteorology; MPI-GE-CMIP6

- GFDL SPEAR Large Ensembles (https://www.gfdl.noaa.gov/spear_large_ensembles/); GFDL-SPEAR-MED

- Energy Exascale Earth System Model (E3SM) large ensembles (publically available: https://aims2.llnl.gov/search/cmip6/?institution_id=UCSB&?experiment_id=historical,ssp370 and https://portal.nersc.gov/archive/home/c/ccsm/www/E3SMv2/FV1/atm/proc/tseries/month_1), obtained for this archive from the NCAR compute systems at */glade/campaign/cgd/ccr/E3SMv1-LE/* and *glade/campaign/cgd/ccr/E3SMv2/*); E3SMv1 and E3SMv2





We ask that this paper and the appropriate references from Table 1 and sources from the list above (for each model used) are
       cited when using the MMLEAv2 data.

## 3    The New CVDPv6

### 3.1    Scope of the package

The original *Climate Variability Diagnostics Package* (CVDP; Phillips et al., 2014) and the related *Climate Variability and
Diagnostic Package for Large Ensembles* (CVDP-LE; Phillips et al., 2020) have been merged into a single application that
incorporates the functionality of both packages. This merged application (henceforth referred to as the *CVDPv6*) is an auto-
mated analysis and graphics tool that facilitates the exploration of modes of climate variability and change in models, including
LEs, and observations. The CVDPv6 computes and displays the major modes of climate variability as well as long-term trends
       and climatologies in models and observations based on a variety of fields including sea surface temperature (sst/tos), surface
       air temperature (tas), sea level pressure (psl), precipitation (pr), sea ice concentration (sic), sea surface height (zos) and the
       Atlantic Meridional Overturning Circulation (AMOC). As an analysis tool, it can be used to explore a wide range of topics
       related to unforced and forced climate variability and change. It can also help with formulating hypotheses, serve as a tool for
model evaluation, and generally facilitate curiosity-driven scientific inquiry.

       When the CVDPv6 is applied to LEs, it computes metrics that are unique to LEs, for example the ensemble-mean (an
       estimate of the forced response) and the ensemble spread due to internal variability. The CVDPv6 operates on a user-specified
       set of model simulations, observational datasets, and time periods, and saves the output (png graphical displays and netcdf data
       files) to a data repository for later access and further analysis. The CVDPv6 also provides the ability to view the output from
two perspectives: *Individual Member* and *Ensemble Summary* (details provided below). A novel feature of the CVDPv6 is the
       option to detrend the data using one of 4 methodologies: linear, quadratic, 30-year high pass filter, and for model LEs, the
       removal of the ensemble mean. This allows for the package to be used as a test-bed for the efficacy of the various observational
       detrending methods in separating the forced response from internal variability, providing important information about how to
       interpret the results when the methods are applied to observations. Another novel feature of the CVDPv6 is that the reference
data set against which the model LEs are compared can be an observational product or the ensemble mean of an individual
       LE or multiple LEs at the users discretion. These novel features of the CVDPv6 could be combined, for example, to show
       the detrended standard deviations of a variable such as surface temperature in future projections with reference to those in
       the historical period, enabling an assessment of forced changes in the characteristics of internal variability. To compare the
       MMLEAv2 archive to the observed historical climate, multiple observational datasets were downloaded and used as reference
data within the CVDPv6 comparisons. The observational reference datasets are listed in Table 2. We note that while this data



**Table 1.** Data included in the MMLEAv2 Dataset. The model and it's reference, forcing type, length of simulation, and variables available are listed in the table. Model names in italics are CMIP6 generation models and forcing, while CMIP5 generation models and forcing are not italicised. The number of ensemble members available for each variable are listed in the table. The variables, scenarios, and set number (sets of LEs we have artificially split the models into for intercomparison purposes) and ensemble size highlighted in italics are those used in the CVDPv6 Tables and Figures. Where an additional variable is listed in brackets the original variable was unavailable and this variable is included in the archive to replace it.

| model CVDP | forcing CVDPv6 set no. (members) | length | sst/tos sea surface temperature | zos sea surface height | pr precip- itation | psl sea level pressure | tas surface air temperature | tauu zonal wind stress | tauv meridional wind stress | zg500 500mb geopotential height | sos sea surface salinity | evspsbl evapo- ration | mlotst mixed layer depth | TXx monthly max dailymaxtas | TNn monthly min dailymintas | Rx1day monthly max pr | siconc/sic sea ice conc. |
|---|---|---|---|---|---|---|---|---|---|---|---|---|---|---|---|---|---|
| *1. ACCESS-ESM1-5* Ziehn et al. (2020) | *hist SSP370/585 Set 1 (40)* | 1850-2100 | 40 | 40 | 40 | 40 | 40 | - | - | 40 | 40 | 40 | 40 | 40 | 40 | 40 | 40 |
| 2. CanESM2 Kirchmeier-Young et al. (2017) | hist RCP8.5 Set 1 (50) | 1950-2100 | 50 | - | 50 | 50 | 50 | 50 | 50 | 50 | 50 | 50 | 50 | 50 | 50 | 50 | 50 |
| *3. CanESM5* Swart et al. (2019) | *hist SSP370/585 Set 1 (25)* | 1850-2100 | 40 25/25 | 65 50/50 | 65 50/50 | 65 50/50 | 65 50/50 | 47 50/50 | - - | 50 50/50 | - - | 40 25/25 | - - | 40 25/25 | 40 25/25 | 40 25/25 | 40 25/25 |
| 4. CESM1 Kay et al. (2015) | hist RCP8.5 Set 1 (40) | 1920-2100 | 40 | 40 | 40 | 40 | 40 | 40 | 40 | 40 | 40 | - | - | 40 | 40 | 40 | 40 |
| *5. CESM2* Rodgers et al. (2021) | *hist SSP370 Set 1 (100)* | 1850-2100 | 100 | 100 | 100 | 100 | 100 | 100 | 100 | 100 | 100 | 100 | 100 | - | 100 | 100 | 100 |
| 6. CSIRO-Mk3-6-0 Jeffrey et al. (2012) | hist RCP8.5 Set 1 (30) | 1850-2100 | 30 | - | 30 | 30 | 30 | 30 | 30 | 30 | 30 | 30 | 30 | 30 | - | 30 | 30 |
| 7. E3SMv1 Stevenson et al. (2023) | hist SSP370 Set 1 (14) | 1850-2100 | 14 | 13 | 14 | 14 | 14 | 14 | 14 | 14 | 13 | 14 | 14 | 14 | 14 | - | 18 |
| 8. E3SMv2 Fasullo et al. (2023) | hist SSP370 Set 2 (21) | 1850-2100 | 21 (TS) | 21 | 21 | 21 | 21 | 21 | 21 | 21 | 21 | 21 | - | 20 | 20 | 20 | 20 |
| *9. EC-Earth3* Döscher et al. (2022) Wyser et al. (2021) | *hist SSP585 Set 3 (8)* | 1850-2100 | 22 58 | - | 22 58 | 22 58 | 23 58 | 23 8 | 23 8 | 22 58 | - | 24 58 | - | 18 58 | 18 58 | 18 58 | 22 58 |
| 10. GFDL-CM3 Sun et al. (2018) | hist RCP8.5 Set 2 (20) | 1920-2100 | - | - | 20 | 20 | 20(TS) | 20 | - | 20 | 20 | - | 19 | - | - | 20 | 20 |
| 11. GFDL-ESM2M Burger et al. (2022) | hist RCP8.5 Set 2 (30) | 1861-2100 | 30 | 30 | 30 | 30 | 30 | 30(U10) | 30(V10) | - | 30 | 30 | 30 | 30 | 30 | - | 30 |
| *12. GFDL-SPEAR-MED* Delworth et al. (2020) | *hist SSP585 Set 2 (30)* | 1921-2100 | 30 | - | 30 | 30 | 30 | - | - | 30 | - | - | - | - | 30 | 30 | - |
| *13. IPSL-CM6A-LR* Boucher et al. (2020) | *hist SSP370/585 Set 2 (6)* | 1850-2100 | 32 11/6 | 33 11/7 | 33 11/7 | 32 11/6 | 33 11/7 | 33 11/7 | 33 11/7 | 33 11/6 | 33 0.8 | 33 0.8 | 33 0.8 | 33 11/6 | 33 11/6 | 33 11/6 | 33 11/6 |
| 14. MIROC6 Tatebe et al. (2019) | hist SSP585 Set 3 (50) | 1850-2100 | 50 | 50 46 | 50 | 50 | 50 | 50 | 50 | 50 | 50 | 50 | 3 | 50 | 50 | 50 | 50 |
| *15. MIROC-ES2L* Hajima et al. (2020) | *hist SSP370/585 Set 3 (10)* | 1850-2100 | 30 10/10 | 31 10/10 | 31 10/11 | 31 10/10 | 31 10/10 | 31 10/10 | 31 10/10 | 30 0/10 | 31 0/11 | 31 0/11 | - - | 30 10/10 | 30 10/10 | 30 10/10 | 30 10/10 |
| 16. MPI-GE Maher et al. (2019) | hist RCP8.5 Set 3 (100) | 1850-2099 | 100 | 100 | 100 | 100 | 100 | 100 | 100 | 100 | 100 | 100 | - | - | - | - | 100 |
| *17. MPI-GE CMIP6* Olonscheck et al. (2023) | *hist SSP370/585 Set 3 (30)* | 1850-2100 | 50 30 | 50 30 | 50 30 | 50 30 | 50 30 | 50 30 | 50 30 | 50 30 | 50 30 | 50 30 | 50 30 | 50 30 | 50 30 | 50 30 | 44 30 |
| *18. UKESM1-0-LL* Sellar et al. (2019) | *hist SSP370/585 Set 3 (13)* | 1850-2100 | 17 16/5 | 17 16/5 | 19 16/5 | 19 16/5 | 19 16/5 | 19 16/5 | 19 16/5 | 19 16/5 | 17 16/5 | 19 16/5 | - - | 16 16/5 | 16 16/5 | 16 16/5 | 17 16/5 |



**Table 2.** List of observational datasets used within the MMLEAv2 CVDPv6 comparisons. Additional datasets not used in the CVDPv6 are included in italics. Note all tas reference datasets are blended products that use surface air temperature over land and sea surface temperatures over the oceans.

| Observational Dataset name | Variables | References |
|---|---|---|
| ERSSTv5 | tos | Huang et al. (2017) |
| HadISST v1 | tos | Rayner et al. (2003) |
| ERA5 | psl, pr, *Rx1day*, *TXx*, *TNn* | Hersbach et al. (2020) |
| ERA20C | psl | Poli et al. (2016) |
| BEST | tas | Rohde and Hausfather (2020) |
| GISTEMP | tas | Lenssen et al. (2019) |
| HadCRUT5 | tas | Morice et al. (2021) |
| GPCC | pr | Becker et al. (2013) |
| GPCP | pr | Adler et al. (2018) |
| NOAA/NSIDC CDR | siconc | Meier (2021) |
| ORAS4 | zos | Balmaseda et al. (2013) |

has been regridded as part of the MMLEAv2, the CVDPv6 does not need input files to be on a common grid for its use as the package does regridding on the fly.

## 3.2 CVDPv6 output

A list of the output from the CVDPv6 can be found in Table 3. A detailed description of each output field is provided in the
Supplemental Materials. All calculations performed in the CVDPv6, including definitions of the modes of variability, are given the *Methodology and Definitions* link at the top of the CVDPv6 output webpage. A number of summary metrics accompany the graphical displays (see Table 2 for a listing). In particular, for each spatial map, the pattern correlation between the model simulation (which could be an individual member of a LE in the *Individual Member* view or the ensemble-mean of a LE in the *Ensemble Summary* view) and the reference data set (typically observations) is given in the upper right of the map. In addition,
for modes of variability based on Empirical Orthogonal Function (EOF) analysis, the fractional variance explained is also provided. The rank maps in the *Ensemble Summary* view, which show the rank of the reference data set within the ensemble spread of the LE, also provide a summary metric of the fractional area of the globe with values between 10% and 90%. For each timeseries displayed in the *Ensemble Summary* view, the ensemble mean and the 25th-75th and 10th-90th percentile ranges across the ensemble members are shown, along with the 10th, 50th and 90th percentile values for linear trends. The percentage
of time that the reference dataset lies within the 10th-90th percentile range of the LE values is also shown. The package also produces a synthesis of model performance based on pattern correlations and RMS errors for 15 key spatial metrics of climate variability (listed in Table 3) as well as an overall benchmark based on a mean score of all 15 metrics combined. These summaries are provided in both graphical and tabular format. A novel feature of the Summary Table is the ability to sort the



**Table 3.** Contents of the CVDPv6. In addition to the information summarised in this Table, the CVDPv6 displays pattern correlations against the reference dataset and the areal fraction of rank values within 10-90% for all diagnostics plotted in map form. It also provides the temporal fraction of rank values within 10-90% and an ensemble mean summary plot for all diagnostics plotted in timeseries form. For the modes of variability computed using EOF analysis, a graphical summary of the distribution of percent variance explained values across the entire ensemble appears above the numerical values of percent variance explained (10th, 50th and 90th).

| Diagnostic | Variables | Season | Plot type |
|---|---|---|---|
| **Summary Metrics** | all | | ensemble table, ensemble graphics, individual table - all for RMS of correlations |
| ENSO | tas | DJF | metric table |
| ENSO | psl | DJF | metric table |
| El Niño Hovmöller | sst | all | metric table |
| El Niño Hovmöller | sst | all | metric table |
| AMV Low-Pass | sst | all | metric table |
| PDV | sst | all | metric table |
| NAO | psl | JFM | metric table |
| PNA | psl | DJF | metric table |
| SAM | psl | DJF | metric table |
| SST std dev | sst | annual | metric table |
| PR std dev | pr | annual | metric table |
| **Climatological Averages** | sst, tas, psl, pr, sic NH, sic SH, zos | seasonal & annual | maps |
| **Standard Deviations** | sst, tas, psl, pr, sic NH, sic SH, zos | seasonal & annual | maps |
| **Linear Trends** | SST, sst, tas, psl, pr, sic NH, sic SH, zos | seasonal & annual | maps |
| **Coupled Modes of Variability** | | | |
| Spatial ENSO | sst, tas, psl, pr, zos | 4 seasons | Composite maps and equatorial Pacific Hovmöller diagrams of El Niño, La Niña and their difference |
| Nino3.4 | sst | monthly | timeseries, standard deviations, spectra, wavelets, autocorrelation, running 30-year standard deviation timeseries |
| AMV, PDV, IPV | sst, tas, pr | monthly | regression mapsm, timeseries, spectra |
| AMOC | sst, tas, overturning | monthly | climatological means, standard deviation, patterns, timeseries, spectra, lag correlations |
| **Atmospheric Modes of Variability** | | | |
| SO, NAM, NAO, SAM, PNA, NPO, PSA, PSA2 | psl | seasonal & annual | timeseries, regression maps |
| **Global Timeseries** | sst, tas, pr, prland | seasonal & annual | timeseries |
| **Regional Timeseries** | | | |
| Various | sst, psl | monthly | timeseries |
| **Sea Ice Extent (NH & SH)** | sie | seasonal, annual, monthly, climatology | timeseries |

values according to a particular metric (for example, the *North Atlantic Oscillation; NAO*) or by the Mean Score across the 15
metrics. The 10th and 90th percentile values for each metric and Mean Score are also provided in the Tables for each model LE, facilitating model performance intercomparisons. The package also produces a range of output listed in Table 3 that can be viewed in *Individual Member* or in *Ensemble Summary* view. The *Individual Member* view gives the option for each ensemble member to be plotted in its native state or as a bias plot compared to a reference dataset (e.g. observations). The *Ensemble Summary* view plots the ensemble mean, the reference dataset, the difference between the two, and the rank of the reference
within the ensemble. If the model and the reference agree well one would expect the reference to sit at a rank inside the model spread i.e. within the 10% and 90% range 80% of the time (Suarez-Gutierrez et al., 2021; Phillips et al., 2020). The rank plot enables a quick evaluation of the variable or metric plotted.

## 3.3 CVDPv6 applied to MMLEAv2

We have run several applications of the CVDPv6 on the MMLEAv2; output can be found at: https://webext.cgd.ucar.edu/ Multi-Case/MMLEA_v2/. In particular, we provide a version without detrending and one where the observations have been quadratically detrended and the LEs have been detrended by removing the ensemble mean as this explicitly removes the forced





response from each LE (note that detrending is omitted from the calculations of climatologies and trends). The CVDPv6 comparisons are divided into 3 sets of models (see Table 1 for which model is in which set) for ease of visualisation. The
CVDPv6 is run on each of the 3 sets of models for the time periods 1950-2022 and 2027-2099. An additional set named Set123 only includes the Summary Metrics (tables and graphics) to facilitate an intermodel comparison across the entirety of the MMLEAv2. All analysis is completed on RCP8.5 and SSP5-8.5 except for GISS-E2-G, CESM2 and UKESM1-0-LL where SSP3-7.0 is used. Observational datasets used for these comparisons are shown in Table 2.

## 4   Intermodel Comparison Tables & Graphics

This section demonstrates the intermodel comparison tables and graphics from the CVDPv6 for all MMLEAv2 ensemble members and types of conclusions that can be drawn from them.

Multiple evaluation methods exist to compare climate variability in models with observations. While they do not always provide consistent answers, together they can be used to create a more general picture of model performance. Here, we use two complementary evaluation methods; pattern correlations (Figure 1) and root mean square differences (RMS; Figure 2).
The pattern correlation informs about the spatial similarity between two maps, while the RMS provides a summary of the relative amplitude of the maps. For the pattern correlation mean score, CESM2 and GFDL-SPEAR have the highest individual ensemble member scores compared to observations (around 0.9), followed by a group of models clustered with their highest correlation in the 0.8-0.85 range. For RMS, the highest performance (lowest RMS values) are found again for GFDL-SPEAR, but also EC-Earth3 and UKESM1-0-LL. These results demonstrate that GFDL-SPEAR is a high performer for both
spatial pattern and amplitude, however, it is worth keeping in mind that the evaluation is also metric-dependent. For example, ACCESS-ESM1-5 has one of the lowest RMS scores and high correlations for ENSO TAS in DJF showing it performs well in both pattern and amplitude for this metric. However, for the La Niña Hovmöller, it is in the bottom two models for the spatial pattern. Another example is GFDL-CM3, which has the lowest pr STD correlation, but conversely performs relatively well on amplitude for the PNA metric. This demonstrates that models can perform well for one metric and evaluation type, but poorly
for another, meaning that one must be careful in selecting an evaluation method and metric fit for purpose for any given study.

There are some metrics where all models tend to perform well or poorly. For example, the red columns of Figure 2, PDV, SST STD and PR STD, all show low error compared to observations in all models and ensemble members for RMS. Conversely, the blue-coloured columns such as ENSO PSL and AMV Low Pass show poor performance across all ensemble members of all models. For the pattern correlation, again all models and members perform poorly for the AMV Low Pass where there is
a maximum correlation of 0.6 (Figure 1). On the contrary, no ensemble members of any model has a correlation below 0.88 for the SAM, highlighting high performance of all models for this metric. Metrics where all models perform poorly can tell us about general biases found in all climate models. The AMV Low Pass is an example of where the models perform poorly in both the spatial pattern and amplitude, athough this could be due to the tendency of the AMV Low Pass index to mix together multiple independent processes (Wills et al., 2019; Deser and Phillips, 2023; O'Reilly et al., 2023) which might have different
relative weights in models and observations. On the flip side the low RMS for SST STD highlights that the amplitude of SST





variability is generally correct in all models, while in many models the maximum correlation is below 0.8 and as low as 0.4 suggesting that while the amplitude is correct there is bias in the spatial pattern. Teasing out where consistent model biases exist and whether they appear for both pattern and amplitude can inform scientific research and model development to improve models in the coming model generations.

Ensemble members from the same model with the same external forcing can have a wide range of performance compared with observations. For metrics such as the AMV Low Pass, the pattern correlation with observations ranges from slightly negative to 0.6 in an individual model. ENSO TAS also has a range of possible correlations that vary by up to 0.3 between members These examples highlight that for metrics such as these two, the use of a single ensemble member would not give a correct model evaluation. For other metrics such as PR STD, the range of correlations within a single model only varies by a

magnitude of 0.1, demonstrating that in this case a single ensemble member is more appropriate for model evaluation. The range of correlations across ensemble members is linked to the magnitude of internal variability of a metric relative to its ensemble-mean (e.g. Milinski et al., 2020; Lee et al., 2021). This range is not determined solely by ensemble size as demonstrated by the SAM metric where the 100-member CESM2 model has a much larger range of possible correlations than the 100-member MPI-GE. This difference in range from two models with the same ensemble size does, however, demonstrate that the variability

of a single metric can be model-dependent similar to Lehner et al. (2020). Another example that highlights the importance of considering the full ensemble, rather than one member is the El Niño Hovmöller, which for CESM2 can have a RMS as small as 0.49 and as large as 0.76. If the member with the 0.76 RMS was used this model would be deemed poor at representing the El Niño evolution, while the opposite conclusion would be made if the 0.49 member was selected. Similar results are again found for smaller ensembles with the PNA in ACCESS-ESM1-5 as low as 0.43 or as high as 0.68 depending on which member

was chosen. In general, Figures 1 and 2 highlight the importance of using LEs for model evaluation, particularly for metrics with high internal climate variability.

## 5 Model Evaluation Figures

This section gives examples of Figures output from the CVDPv6 and associated insights that can be made into the MMLEAv2 models.

The CVDPv6 can be used to assess annual global average surface temperature and determine whether the response to external forcing (i.e. greenhouse gases, anthropogenic aerosols and volcanic forcing) in combination with internal variability is similar to observations. For the Set 1 MMLEAv2 models shown in Figure 3, only CESM2 and CESM1 largely encompass the observations within the model spread as highlighted by the temporal ranks (e.g. the percentage of time that the observed value lies within the 10th-90th percentile range of the model's ensemble members shown in the bottom left of the plots), where

these models have 85% and 79% respectively. ACCESS-ESM1-5, CanESM2 and CanESM5 all have a larger climate sensitivity than is estimated from observations, as demonstrated by all members warming much more than the observations at the end of the historical period. The CSIRO-Mk360 model has a bias in the historical period where it does not warm from 1960-2000, dissimilar to what is observed. In response to volcanic forcing, all models have a distinctive dip in temperature in 1963 (Agung),



1982 (El Chichón), and 1991 (Pinautubo) that is similar in magnitude to the observed record, except CSIRO-Mk36 which does
dip in response to eruptions but at a different rate to observations. This suggests that the global-mean annual-mean temperature
response to large tropical volcanic eruptions is realistic in most Set 1 models.

Accurately representing large-scale modes of climate variability in models is important as these modes are key components
of the climate system. The large-scale modes shown in the following examples are calculated after the removal of the ensemble
mean. Hence they approximate pure internal variability with all external trends removed. This is a key use of LEs and a
powerful tool within the CVDPv6. We note that changes in the variability itself are not removed using this methodology, so
changes in the variability itself can be assessed using this method. The CVDPv6 uses a rank histogram approach where the
rank of observations is shown within each model's spread. This type of comparison between observations and models is only
possible using LEs. The NAO pattern in DJF for the Set 1 models is shown in Figure 4. CESM2 and CanESM5 have the largest
percentage of white area (indicating that observations lie in the center of the distribution of the model's diagnosed internal
variability patterns) in the rank histogram map (areal percentage of observed values lying within the 10th-90th percentile of the
LE values of 61% and 59% respectively), while CSIRO-Mk36 has the lowest at 28%. For the most part, the Set 1 models tend
to have similar biases with the NAO variability overestimated in the North Pacific and over Alaska, and underestimated in the
North Atlantic and over most of Russia and Siberia. A similar plot for the PDV is shown in Figure 5 for Set 3 models. In this
case, the pattern biases are quite different for different models, especially in the tropical Pacific, where some models have too
much PDV variability and some have too little. There is a consistent bias in the North Pacific, where all Set 3 models show too
large Kuroshio-Oyashio Extension anomalies that are located too far north. EC-Earth3, MPI-GE and MPI-GE-CMIP6 are the
models with more than 60% of the areal percentable of the ranks occurring between the 10th and 90th percentile, indicating
that they capture observed PDV best out of the Set 3 models. Modes of variability can also be assessed by their spectra (Figure
6; ENSO spectra). For the ENSO spectra in Set 2 models, we find that GFDL-CM3, E3SMv2 and IPSL-CM6A have a peak that
sits at 3 years, while observations have a broader 3-7 year peak. GFDL-ESM2M is similar to observations but with a stronger
peak as is GISS-E2-G with an even stronger peak. GFDL-SPEAR and E3SMv1 however, are much more in agreement with
observations, with the ensemble spread encompassing observations.

Another important aspect of model evaluation is the consideration of how well a model represents the observed impacts of
a mode of variability. In Figure 7 we demonstrate the SON precipitation impacts from La Niña in the Set 3 models. MIROC6
has 70% of the areal percentable of ranks between the 10th and 90th percentile, with all other models above 62%. This shows
that over 60% of the globe has rainfall impacts in SON that are well represented in the Set 3 models. Be that as it may,
biases do exist. For example, all models have a wet bias in the western tropical Pacific, and the Alaskan coastline and a dry
bias over California in relation to observed La Niña events. In other regions models have differing biases with the La Niña
teleconnection to Australia overestimated in some models and underestimated in others. Whether these biases are due to model
errors or uncertainty in the observed composite due to limited sampling of events (Deser et al., 2018, 2017) and/or data issues is
less clear due to a sparsity of observations in some regions. While not shown in this paper, the CVDPv6 allows for comparison
with multiple observational products, which can help characterize observational uncertainty.



While we cannot directly compare future projections with observations, we can compare models with each other and with the multi-model or in this case multi-ensemble mean. The CVDPv6 outputs data in netcdf files, which can then be used to create new plots. We created Figure 8 from these outputs. It shows (for Set 2 models) an individual model comparison to the multi-ensemble mean (MEM; calculated across all models from the 3 sets). This method of model intercomparison highlights where the forced response differs across models. We find that the trend in temperature in GFDL-ESM2M is smaller than the MEM, IPSL-CM6A and E3SMv1 have a larger trend than the MEM, and GFDL-CM3 has a larger trend everywhere but Antarctica and the Southern Ocean where it is smaller. E3SMv2 and GFDL-SPEAR have similar trends to the MEM. In this case, differences from the MEM largely show differences in global warming between models, due to differences in climate sensitivity and emissions scenario, but a similar comparison for other variables would help to characterize other aspects of the structural uncertainty in climate projections. Interestingly there are substantial regional changes in the rank of each model within the ensemble spread. While both E3SMv1 and v2 sit at high ranks in the higher latitudes, and GFDL-ESM2M sits at low ranks for all regions, IPSL-CM6A sits at a high rank in parts of the extratropics and low in the tropics while GFDL-CM3 has low ranks only in the Southern Ocean and GFDL-SPEAR has low ranks in specific locations in the extratropical northern hemisphere. This means that regional warming is not solely dependent on the magnitude of global warming in these models. We note that this type of comparison could also be done at warming levels which would present a fairer comparison across the varying future scenarios. This is, however, not possible with the CVDPv6 package alone.

To highlight the use of the extreme indices available in the MMLEAv2 dataset we show the change at the end of the century (2090-2099) compared to the historical period (1950-1959) of the monthly maximum of daily maximum temperature (TXx) in June, July, August (Figure 9). For a fairer comparison of the varying future scenarios (compared to Figure 8) we only use 8 models that have the SSP370 future scenario available for this variable. In the SSP370 future scenario, all models show an increase in TXx over land (and over almost all of the ocean). The magnitude of this change is, however, model dependent. E3SMv1 shows the largest increase followed by UKESM1-0-LL, with MPI-GE-CMIP6 and MIROC-E3SL showing the lowest increases. The range of increases in TXx over land is greater than $10^{o}$C, highlighting how variable the future projections of this extreme metric are across LEs. Figure 3b in (Deser et al., 2020) reported a similar finding for daily July heat extremes at the grid box containing Dallas Texas.

## 6 Conclusions

This work has presented two new complementary resources for the study of climate variability and change: the MMLEAv2 and the CVDPv6. Designed for ease-of-use, these tools provide a broad synthesis of internal and forced contributions to the major modes of climate variability and trends in model LEs in relation to observations, facilitating model evaluation, and inter-comparison. Here, we have demonstrated some of the insights that can be derived by applying the CVDPv6 to the MMLEAv2. In particular, we have highlighted the following points:

– the MMLEAv2 is an extension of the MMLEA (Deser et al., 2020) with additional models and variables available (18 models and 15 two-dimensional variables)



- – the MMLEAv2 is remapped onto a common 2.5 x 2.5 degree grid for ease of inter-model comparison

- – The MMLEAv2 also provides observed reference datasets on the common 2.5 x 2.5 degree grid for ease of model-to-observation comparison

- – The CVDPv6 provides a powerful and efficient way to analyse LE output, including the MMLEAv2

- – A preliminary model evaluation has been completed using the CVDPv6 applied to MMLEAv2 that highlights common model biases, and which models perform well or poorly compared to observations

- – The CVDPv6 enables the exploration of climate model output and observations, including modes of variability, time-series, trends, and spatial maps of key variables with multiple detrending methods

- – The CVDPv6 can use any reference dataset (model or observations) and time period for comparison

- – the CVDPv6 output is provided in two complementary ways: *Individual Member* view and in *Ensemble Summary* view

- – *Ensemble Summary* view includes not just spatial plots and timeseries, but also a rank plot of where the observations sit within the model spread for easy evaluation (i.e. white areas show high model-to-observational agreement)

- – Netcdf output is also available from the CVDPv6, which allows uses to create their own additional figures from diagnostics computed in the package.

- – Graphical output is available from the CVDPv6 which can be output as png files for publications and presentation

- – Output is available as an easily navigable HTML format for users to click through (i.e. https://webext.cgd.ucar.edu/Multi-Case/MMLEA_v2/)

Additionally, we have demonstrated the utility of LEs for model evaluation. Observations can be considered as one realisation of the world that we live in, meaning that observations are best compared with multiple realisations from a climate model, such
as a LE. This is highlighted by the spread of evaluation metrics found across the ensemble. In some cases one ensemble member would evaluate poorly against observations while another would compare favourably. For this reason LEs are important for model evaluation, especially for highly variable quantities. Overall, the MMLEAv2 will allow for exciting new science using LEs and the CVDPv6 is a powerful tool made specifically for analysis of LEs and their unique characteristics.

*Code and data availability.* The CVDPv6 code is available at: https://www.cesm.ucar.edu/projects/cvdp/code The MMLEAv2 data is cur-
rently in the process of being made available via the MMLEA website and the NCAR Research Data Archive: https://www.cesm.ucar.edu/community-projects/mmlea and https://rda.ucar.edu/. The data will be fully available on final publication of this manuscript. All processed data, and a frozen version of the CVDPv6 code used in this publication are available on zenodo at doi:10.5281/zenodo.14292580 https://zenodo.org/records/14292580



*Author contributions.* NM led the compilation of the MMLEAv2 and wrote the manuscript. AP led the CVDPv6 Package Development and
ran the package on the MMLEAv2 data. CD was key in the intellectual development of the CVDPv6. NM, AP & CD designed the manuscript.
RW was key in adding data to the MMLEAv2 archive and initiating it's existence. FL compiled the MMLEA original archive. JF is a CVDP
super-user who has helped develop the CVDPv6 over many years. JC specifically helped with the testing and development of CVDPv6. LB
& UB set up the cmip-ng archive at ETH that provided much data for the MMLEA. All authors contributed to the revision of this manuscript.

*Competing interests.* The authors declare no competing interests are present.

*Acknowledgements.* We would like to acknowledge computing support from the Casper system (https://ncar.pub/casper) provided by the
NSF National Center for Atmospheric Research (NCAR), sponsored by the National Science Foundation. This research was supported by
the Australian Research Council Discovery Early Career Researcher Award DE230100315. Nicola also acknowledges support from NASA
(grant 80NSSC23K0358) and the CIRES Visiting Fellows Program and the NOAA Cooperative Agreement with CIRES, NA17OAR4320101.
We thank C2SM for providing/updating cmip6-ng and Ruth Lorenz for providing the zos fields as part of this archive. We also thank Dillon
Amaya for his work in quality-checking the datasets. We additionally thank Thomas Frölicher and Dirk Olonscheck for providing data and
access for GFDL-ESM2M and MPI-GE-CMIP6 respectively. The National Center for Atmospheric Research (NCAR) is sponsored by the
National Science Foundation under Cooperative Agreement 1852977. The efforts of Drs. Fasullo, Caron, and Lehner are supported by the
Regional and Global Model Analysis (RGMA) component of the Earth and Environmental System Modeling Program of the U.S. Department
of Energy's Office of Biological & Environmental Research (BER) under Award Number DE-SC0022070. Dr. Fasullo was also supported
by NSF Award 2103843. RCJW was supported by the Swiss National Science Foundation (Award PCEFP2 203376). LB has been funded by
the Deutsche Forschungsgemeinschaft (DFG, German Research Foundation) under Germany's Excellence Strategy – EXC 2037 "CLICCS –
Climate, Climatic Change, and Society" – project no. 390683824, a contribution to the Center for Earth System Research and Sustainability
(CEN) of the University of Hamburg.



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



**Figure 1.** Pattern correlation of each ensemble member of all MMLEAv2 models with the first specified observational dataset (for each variable for each metric) is compared to every other specified observational dataset and every model simulation in this figure directly from the CVDPv6. Shown for multiple modes of variability, the standard deviation of sst and pr, and a mean score across all metrics used in the Figure (after applying a Fischer z-transform). Computations are completed over the period 1950-2022. Observations are detrended using a quadratic fit, while LEs are detrended by removing the ensemble mean.





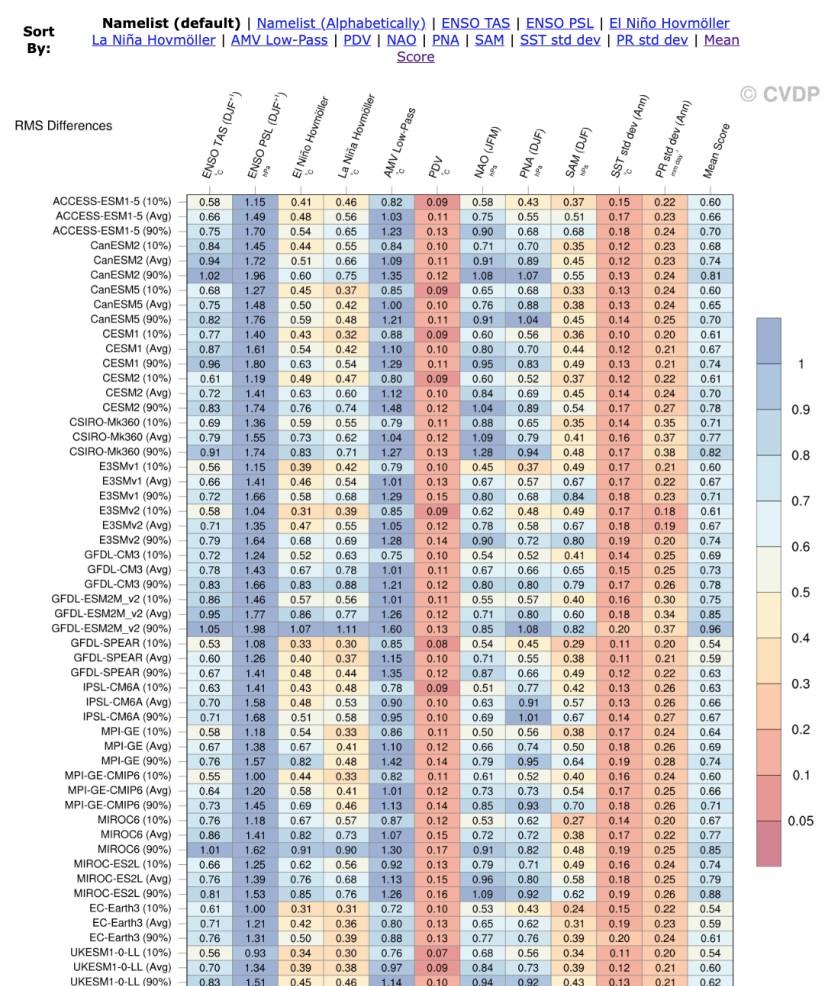

**Figure 2.** RMS difference between each MMLEAv2 model ensemble average and observations as well as the 10th and 90th percentile of the RMS difference across the ensemble. Shown for multiple modes of variability, the standard deviation of sst and pr, and a mean score across all metrics used in the Figure (after normalizing each by the spatial RMS of the observed pattern to account for the different units of each variable). Computations are completed over the period 1950-2022. Observations are detrended using a quadratic fit, while LEs are detrended by removing the ensemble mean. Area-weighted pattern correlations and RMS differences are calculated between observations and each model simulation (regridded to match the observational grid) for 11 climate metrics. The Total Score column shows the average of the 11 pattern correlations (Z-transformed) and RMS differences. The following domains are used to compute the pattern correlations and RMS differences: Means, standard deviations, ENSO and PDV: Global; AMV: 63°S:65°N; El Nino and La Nina Hovmöllers: entire longitude/temporal range shown, NAM/NAO (20:90°N) and SAM (20:90°S).





**Figure 3.** Global average surface air temperature anomalies for Set 1 computed annually over the period 1950-2022. The dark blue curve shows the model's ensemble mean timeseries, and the dark (light) blue shading around this curves depicts the 25th-75th (10th-90th) percentile spread across the ensemble members. Observations are shown in the thick gray curve with the dataset name and trend of the period written in gray under the Figure title. The last panel shows the ensemble mean of each LE as well as observations. The 10th, 50th and 90th percentile values of the linear trends across the model are shown in the top right of each panel and the percentage value on the bottom left of each panel is the percentage of time that the observed value lies within the 10-90th percentile of the LE values. Here, each vertical bar denotes a different ensemble member, and the 10th, 50th and 90th percentile values are identified with taller bar. Set 2 & 3 can be found here: https://webext.cgd.ucar.edu/Multi-Case/MMLEA_v2/MMLEA_Set2_nonenone_1950-2022/tas_global_avg_ann.summary.png & https://webext.cgd.ucar.edu/Multi-Case/MMLEA_v2/MMLEA_Set3_nonenone_1950-2022/tas_global_avg_ann.summary.png



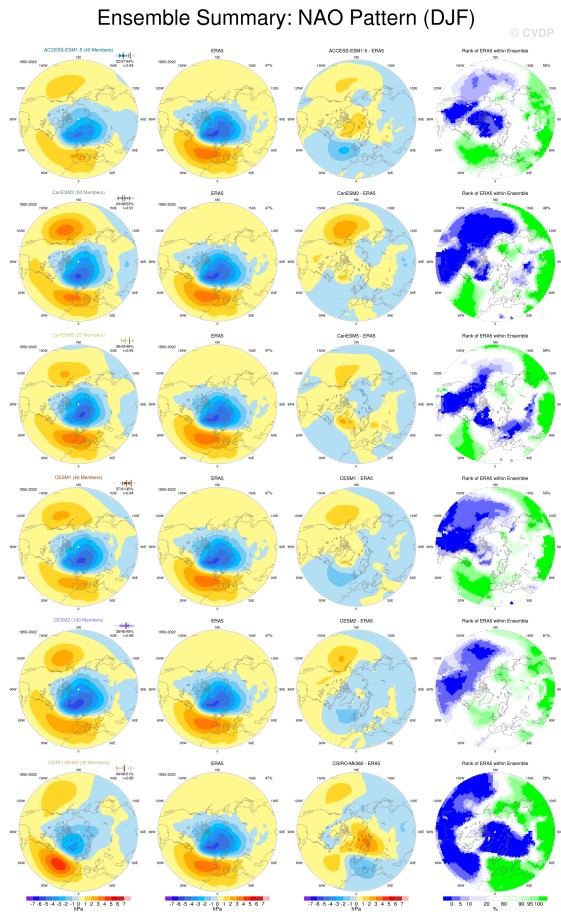

**Figure 4.** The North Atlantic Oscillation (NAO) Pattern in DJF for: left) ensemble mean of Set 1 models, middle left) observations, middle right) difference between the ensemble mean and observations and right) rank of observations within the ensemble spread. The percent variance explained (PVE) by the NAO over its native (EOF) domain is given in the subtitle at the top right of the regression map: the first, second and third values indicate the 10th, 50th and 90th percentile values across the ensemble, respectively. A graphical summary of the distribution of PVE values across the entire ensemble appears above the numerical values of PVE: each vertical bar denotes a different ensemble member, and the 10th, 50th and 90th percentile values are identified with taller bars. The observed PVE value is marked by a gray vertical bar. To quantify the degree of resemblance between the simulated and observed NAO regression maps, a pattern correlation (r) is computed between the observed NAO regression map and the ensemble average of the simulated NAO regression maps (e.g., maps in columns 1 and 2) over the domain shown. This r value is displayed at the upper right of each model panel, just below the range of PVE values White areas on the observed percentile rank maps indicate regions where the observed value lies within 10-90% of the model LE values, indicating the model is likely to be realistic. The value to the right of each rank map denotes the areal percentage of observed values lying within the 10th-90th percentile of the LE values. Computations are completed over the period 1950-2022. The NAO is the leading EOF in the region [20-80N, 90W-40E] following Hurrell and Deser (2009). Observations are detrended using a quadratic fit, while LEs are detrended by removing the ensemble mean. Set 2 & 3 can be found here: https://webext.cgd.ucar.edu/Multi-Case/MMLEA_v2/MMLEA_Set2_quadrmEM_1950-2022/npo_pattern_djf.summary.png & https://webext.cgd.ucar.edu/Multi-Case/MMLEA_v2/MMLEA_Set3_quadrmEM_1950-2022/nao_pattern_djf.summary.png





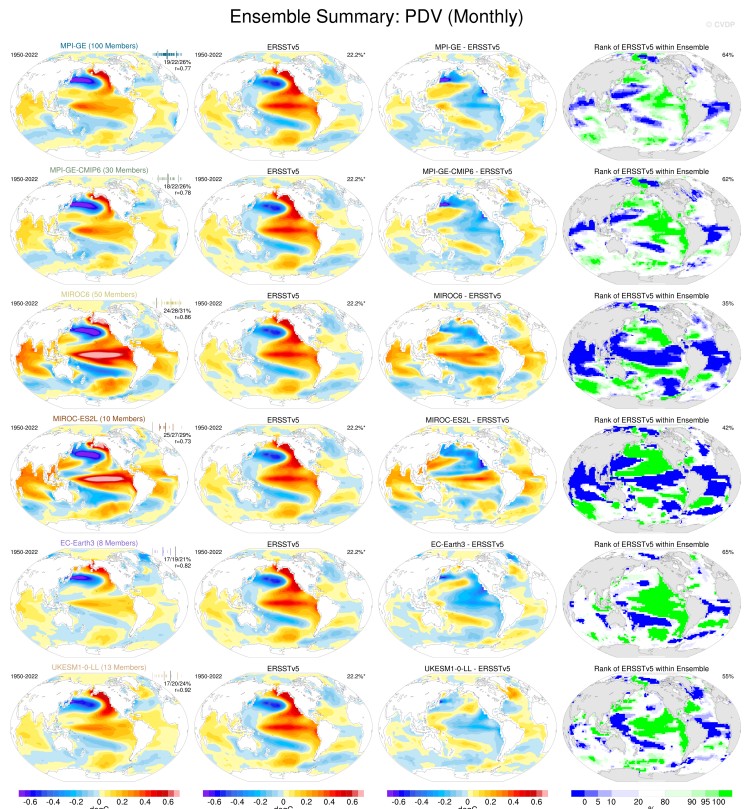

**Figure 5.** The Pacific Decadal Variability (PDV) Pattern in DJF for: left) ensemble mean of Set 3 models, middle left) observations, middle right) difference between the ensemble mean and observations and right) rank of observations within the ensemble spread. The percent variance explained (PVE) by the PDV over its native (EOF) domain is given in the subtitle at the top right of the regression map: the first, second and third values indicate the 10th, 50th and 90th percentile values across the ensemble, respectively. A graphical summary of the distribution of PVE values across the entire ensemble appears above the numerical values of PVE: each vertical bar denotes a different ensemble member, and the 10th, 50th and 90th percentile values are identified with taller bars. The observed PVE value is marked by a gray vertical bar. To quantify the degree of resemblance between the simulated and observed PDV regression maps, a pattern correlation (r) is computed between the observed PDV regression map and the ensemble average of the simulated PDV regression maps (e.g. maps in columns 1 and 2) over the domain shown. This r value is displayed at the upper right of each model panel, just below the range of PVE values White areas on the observed percentile rank maps indicate regions where the observed value lies within 20-80% of the model LE values, indicating the model is likely to be realistic. The value to the right of each rank map denotes the areal percentage of observed values lying within the 10th-90th percentile of the LE values. Computations are completed over the period 1950-2022. The PDV Index is defined as the standardized principal component (PC) timeseries associated with the leading Empirical Orthogonal Function (EOF) of area-weighted monthly SST anomalies over the North Pacific region [20-70N, 110E-100W] minus the global mean [60N-60S] following Mantua et al. (1997). Observations are detrended using a quadratic fit, while LEs are detrended by removing the ensemble mean. Set 1 & 2 can be found here: https://webext.cgd.ucar.edu/Multi-Case/MMLEA_v2/MMLEA_Set1_quadrmEM_1950-2022/pdv.summary.png & https://webext.cgd.ucar.edu/Multi-Case/MMLEA_v2/MMLEA_Set3_quadrmEM_1950-2022/pdv.summary.png





# Ensemble Summary: Niño3.4 SST Power Spectra (Monthly)

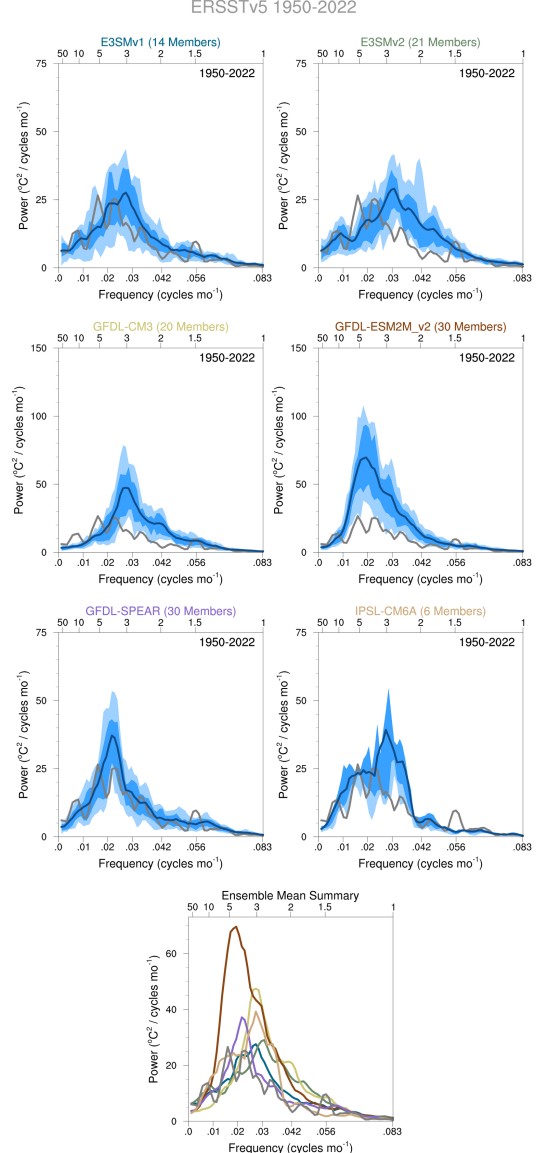

**Figure 6.** ENSO (nino3.4) monthly power spectrum for Set 2 models. The dark blue curve shows the model's ensemble mean time-series, and the dark (light) blue shading around this curves depicts the 25th-75th (10th-90th) percentile spread across the ensemble members and the grey line is the observations. The bottom right shows the ensemble mean from each model and the observations on a single panel. Computations are completed over the period 1950-2022. Power spectra are computed in variance preserving format from the linearly detrended December Nino3.4 SST Index (SST anomalies averaged over the region 5N-5S, 170W-120W). Observations are detrended using a quadratic fit, while LEs are detrended by removing the ensemble mean. Set 1 & 3 can be found here: https://webext.cgd.ucar.edu/Multi-Case/MMLEA_v2/MMLEA_Set1_quadrmEM_1950-2022/nino34.powspec.summary.png & https://webext.cgd.ucar.edu/Multi-Case/MMLEA_v2/MMLEA_Set3_quadrmEM_1950-2022/nino34.powspec.summary.png







**Figure 7.** Composite of precipitation in SON for La Niña events for: left) ensemble mean of Set 3 models, middle left) observations, middle right) difference between the ensemble mean and observations and right) rank of observations within the ensemble spread. The top right of the rightmost panels shown the percentage of the map that is white, where observations are considered to be within the ensemble spread. Computations are completed over the period 1950-2022. The pattern correlations are displayed at the lower right of each model panel. The number of events that go into each spatial composite are displayed at the upper right of each panel (given as an average per ensemble member and as a total over all ensemble members). The value to the right of each rank map denotes the areal percentage of observed values lying within the 10th-90th percentile of the LE values. Observations are detrended using a quadratic fit, while LEs are detrended by removing the ensemble mean. Set 1 & 2 can be found here: https://webext.cgd.ucar.edu/Multi-Case/MMLEA_v2/MMLEA_Set1_quadrmEM_1950-2022/nino34.spatialcomp.lanina.pr.summary.son0.png & https://webext.cgd.ucar.edu/Multi-Case/MMLEA_v2/MMLEA_Set2_quadrmEM_1950-2022/nino34.spatialcomp.lanina.pr.summary.son0.png



**Figure 8.** Annual surface air temperature trend from 2027-2099 for: left) ensemble mean of Set 2 models, middle left) multi-ensemble mean (MEM), middle right) difference between the ensemble mean and the MEM and right) rank of the model ensemble mean within the MEM. All analysis is completed on RCP8.5 and SSP5-8.5 except for GISS-E2-G,CESM2 and UKESM1-0-LL where SSP3-7.0 is used.



**Figure 9.** Monthly maximum of the daily maximum temperature (TXx) averaged over June-July-August in the 8 models that have data for the SSP370 future scenario. left) Individual model ensemble mean for the period 1950-1958, middle left) the change in 2090-2099 as compared to 1950-1958, middle right) the change in 2090-2099 as compared to 1950-1958 in the multi-ensemble mean (MEM) of the 8 models shown on the plot, right) change in the single model ensemble mean minus the MEM change.