# Peer review of "The updated Multi-Model Large Ensemble Archive and the Climate Variability Diagnostics Package: New tools for the study of climate variability and change"

_EGUsphere, 2024_

## Author Response (AR1)

We thank the editors and reviewers for their time in reviewing this manuscript. All line numbers reference in our response refer to the track changed version for clarity.

**Response to Editor comments below:**

1. Climate model code would need to be provided. This has now been further clarified and this is not required.

Thank you for clarifying this point.

2. The embargo on MMLEAv2 data. The confusion has come about due to having the full MMLEAv2 dataset that is not yet hosted (you have embargoed it) and the additional data which has been provided in the zenodo repo (at my request to comply with GMD's policy) in order to reproduce the results on the manuscript. I believe there was confusion about what was in the zenodo repo and what is pending. I would like to suggest that you publish your data and update the data availability to reflect this. Could you also comment on if any permanent data archiving is planned or any way of version control or data DOIs may be managed?

We apologise for the delay. The data has been submitted for permanent publication (which can include a DOI) we were advised it would be online in early January, but there have been substantial delays. The data is now available at https://rda.ucar.edu/datasets/d651039/. The data is hosted on NCAR's Research Data Archive, a long term, publicly available platform for data storage, where many climate related datasets are hosted.

3. The reference/sources to model data are not yet complete. The manuscript needs to provide sufficient information so that someone reading your paper could reproduce your dataset and results. Are their persistent DOI's, data citations or similar you could include? You mention data from two colleagues, could you expand on these runs please? How would someone get the data, is it available, would someone need to contact the author etc (anything relevant really)? Please note that the following link does not work https://esgf-data.dkrz.de/projects/mpi-ge

We have replaced the data sources with where the permanent data storage for each model is located including a reference to the dataset paper (same as in the table) and a reference to the doi for each model (where available). Please see line 102 onwards for the new list of data sources.

**Response to Reviewer 1:**

The manuscript documents the updated multi-model large ensemble archive and the climate variability diagnostics package version 6 (CVDPv6). Examples of different analyses available with CVDPv6 are also provided.

Overall the paper provides a reference for two extremely valuable resources and should be published. However, the analysis highlighted does not stress the need to critically evaluate the fidelity of the modelled forced response. In particular, potential signal to noise errors in the forced response would directly impact the validity of the modelled internal variability, with major implications for any comparison to observations. The multi-model large ensemble archive is a key tool to make progress here and I recommend adding some discussion on this.

We agree with the Reviewer that it is critical to evaluate the fidelity of the modelled forced response and the fidelity of the modelled internal variability. However, isolating the forced response in observations, and isolating internal variability in observations, remain active areas of research and there is no "best practice" at this point. It is well beyond the scope of the CVDPv6 to provide such information. Ongoing efforts such as the "Forced Component Estimation Statistical Method Intercomparison Project" (ForceSMIP), which is co-led by two of the authors of this paper (R. Wills and C. Deser), (https://sites.google.com/ethz.ch/forcesmip/about?authuser=0) are expected to provide guidance on how best to extract forced response and internal variability from the observational record. We note that the MMLEA is being used as a methodological testbed in ForceSMIP, further highlighting its value as a key tool for making progress on this critical issue. We hope to use the results from ForceSMIP in future versions of the CVDP.

To this point we have added text on line 33 reading "*This large ensemble size allows for the assessment of not just the mean state of a model, but its internal variability and forced response in combination in comparison to observations (e.g. Suarez-Gutierrez et al, 2021), a comparison that has become possible with the lengthening observational record and increasing forced trends (Simpson et al, 2025)*" as well as on line 321 "*While such analysis allows comparison between models it does not enable us to assess which model is most realistic. Research using emergent constraints from observations could be applied to the MMLEAv2 to answer such a question.*"

Line 19: state that it is the modelled forced response and internal variability that can be quantified - the real world could be very different.

This has been changed to read "*A single LE allows for both the quantification and separation of the modelled forced response (response of any given variable to external forcing) and unforced internal variability*" on line 20.

Line 29: evaluating whether the observations sit within the model spread is insufficient - the fidelity of the model response to external forcings (and relatedly its representation of internal variability) is crucial.

We have added the following statement to the text '*This large ensemble size allows for the assessment of not just the mean state of a model, but its internal variability and forced response in combination in comparison to observations (e.g. Suarez-Gutierrez et al, 2021), a comparison that has become possible with the lengthening observational record and increasing forced trends (Simpson et al, 2025)*' on line 33

Line 74: I believe there are several other LEs in the CMIP6 archive. at least over the historical period - are there plans to include these?

We made the choice to add any LEs we could obtain the data to from CMIP6 that has historical + ssp370 or ssp585, and correspondingly, some LEs that only cover the historical period were excluded. We do not plan to add additional LEs at this point.

Line 85: including winds at different levels would be a great addition - most extremes are related to atmospheric circulation and winds are crucial for assessing the forced response and internal variability of extremes.

This would indeed be useful as would some other variables. We have chosen to make the archive public now to enable scientific research with what we have rather than spend more time adding additional variables. We note that this is a community effort that relies on scientists to process the data. If possible we will continue to add variable such as the ones suggested here to the archive post-publication.

Line 138: detrending does not separate external forcing and internal variability since some external forcing including aerosols, solar and volcanoes, produce variability beyond the trend.

We have modified the text to read '*A novel feature of the CVDPv6 is the option to detrend the data using one of 4 methodologies: linear, quadratic, 30-year high pass filter, and for model LEs, the removal of the ensemble mean. While removing the ensemble mean effectively removes the response to external forcing, the other methodologies are only effective at removing the long-term trend. The inclusion of multiple methods allows for the package to be used as a test-bed for the efficacy of the various observational detrending methods in separating the forced response from internal variability, providing important information about how to interpret the results when the methods are applied to observations.*' on line 155

Line 143: please add 'modelled' to 'internal variability'.

Modelled has been added on line 165

Line 151: could you provide a link to the CVDPv6 output webpage please?

This has been added on line 179

Line 152: should this say Table 3?

Yes – this has been corrected.

Line 153: would that be the average pattern correlation for individual members?

No, it is the pattern correlation for the ensemble-mean map shown in the figure. This should be similar to the average pattern correlation for individual members, but they would have to be Z-transformed before averaging.

Line 175: please check the link - it didn't work for me

Apologies – it works but the website went offline for a week for maintenance and this likely when you tried to access it. It is now accessible again.

Figure 1: apologies I don't understand this, what does a pattern correlation mean for an index such as ENSO which is a timeseries? Also, are all panels needed or would it be better to illustrate with one or two?

This is the ENSO pattern – i.e. the spatial Pattern in the tropical Pacific. All of the metrics shown in Figs. 1 and 2 are spatial patterns, not timeseries.  For example, "ENSO TAS (DJF+1)" is the spatial map of TAS anomalies in DJF+1 based on ENSO (e.g., El Nino minus La Nina) composites.  This has been clarified in the revised figure caption, which now reads *"Note that all of the metrics shown in Figs. 1 and 2 are spatial patterns.  For example, "ENSO TAS (DJF+1)" is the spatial map of TAS anomalies in DJF+1 based on ENSO (e.g., El Nino minus La Nina) composites, "PR std dev (Ann)" is the map of PR standard deviation based on annual means, and "El Nino Hovmoller" is the Hovmoller diagram of El Nino composites of Equatorial Pacific SST anomalies over the*

*longitude domain 120E-80W and time domain Jan year 0 – May year+2 (see Methodology and Definitions link in the CVDPv6)."*

Figure 2: same comment as Fig 1, also what are the colours?

The colours are the RMS value. The caption has been updated to clarify this and now reads '*RMS difference between each MMLEAv2 model ensemble average and observations as well as the 10th and 90th percentile of the RMS difference across the ensemble (shown in both colours and numbers in the table).*'

Line 220: but presumably evaluating more members is better than a single member?

Yes – but perhaps not necessary. The text has been updated to read '*demonstrating that in this case a large ensemble is not necessary for model evaluation*' on line 248.

Figure 3: Please provide a legend for the ensemble summary panel

The colours come from the titles in the plots above this is clarified in the caption which now reads '*The last panel shows the ensemble mean of each LE (colour codes by the colour of each model's title in the previous panels) as well as observations.*'

Figure 4: There are far too many panels and the text is impossible to read. I suggest just showing a couple of models to illustrate the type of output. It appears that the model patterns are based on the ensemble means which should not be compared to observations - if this is not the case please clarify. The observations are the same in every row so really do not need to be repeated.

This is already a subset of models and our preference is to keep it as is so that the reader can compare multiple models and see the output as it is displayed in the CVDPv6.

Lines 247-267: This ignores potential impacts of forcings on the modes that could be very important. I am certain many of the authors are aware of this, indeed I believe the leadrecieved author has written about volcanic impacts on ENSO, so I am puzzled why this is ignored here.

Line 250 already addresses this point by saying that '*We note that changes in the variability itself are not removed using this methodology, so changes in the variability itself can be assessed using this method.*' we have updated this on revision in response to your comment to read '*We note that changes in the variability itself due to external forcing are not removed using this methodology*' on line 278.

Figure 5: This appears to illustrate the same output as Fig 4 so could be removed

Figure 5 is the PDV while Figure 4 is the NAO so they show different modes of variability.

Figure 6: I presume the dark blue curve shows the model power spectrum rather than the timeseries? Again, I think the average of ensemble members rather than the ensemble mean should be compared with observations. The caption states that observations are grey but it looks more like brown to me.

The dark blue curve shows the average of the power spectra computed for each ensemble member. This has been clarified in the revised caption.

Line 271: also worth pointing out that La Nina rainfall impacts are not well represented over up to 40% of the globe. Is there a way to highlight, perhaps with stippling, where the observations lie outside the range of ensemble members?

This is already shown in the ranks on the RHS.

Figure 8: GISS-E2-G,CESM2 and UKESM1-0-LL are not shown

This is because they are not in set2 models (see caption).

Lines 278-293: Highlighting model differences is very important, but I suggest also highlighting the need to address these differences perhaps through emergent constraints.

We have added the following text '*While such analysis allows comparison between models it does not enable us to assess which model is most realistic. Research using emergent constraints from observations could be applied to the MMLEAv2 to answer such a question.* ' on line 321

Figure 9: the period in the caption (1950-1958) is not consistent with the plots (1950-1959). Again, I suggest showing fewer models to illustrate the output more clearly.

As previously we would like to keep all models here, the caption is incorrect has been updated to match the plots.

Line 296: Can CVDPv6 also compare different scenarios?

Yes, the CVDP can compare different scenarios (across different times) in the same comparison. The CVDP can read in data from any simulation if the data follows CMIP's file convention format.

**Response to Reviewer 2:**

**General comments**

The authors present a valuable contribution to the field of climate model evaluation, the Climate Variability Diagnostics Package (CVDP), which is developed for collective evaluation of simulated climate variability modes. The manuscript is well-structured, and the availability of CVDP results for the Multi-Model Large Ensemble Archive (MMLEA) alongside the tool itself represents an important resource for the community. I appreciate the authors' efforts in developing CVDPv6 and see its value. However, it would be helpful to clarify what is novel beyond the integration of CVDP and CVDP-LE. Does CVDPv6 introduce any new scientific advancements or methodological improvements? Highlighting these aspects in a more clear way would further strengthen the manuscript.

Thank you for your positive feedback. There are indeed methodological improvements such as the inclusion of multiple detrending methods in the CVDPv6.

This is highlighted in two additional conclusions added to the text:

'*Multiple detrending methods can be applied in the CVDPv6, an improvement on previous versions of the package allowing for the efficacy of methods such as quadratic detrending that are typically applied to single realisations (such as observations) to be tested against the removal of the ensemble mean approach (effectively removing effects of external forcing)*' on line 351

'*Figures 1 & 2 of this paper allow the user to determine which climate variables need a large ensemble for fair assessment again observations*' on line 347 and "*By leveraging the combination*

*of the CVDPv6 and MMLEAv2 presented in this paper we can determine which climate variables need a large ensemble for fair assessment against observations."* in the abstract.

For the CVDP the following improvements on previous versions are also added and now highlighted on line 169

1) Addition of a new IPV Index based on Henley et al. (2015)

2) The PDV and AMV definitions have been updated so that the global-mean SST anomaly is no longer removed following Mantua et al. (1997) and Deser and Phillips (2023), respectively.

3) SST, TAS, PR and ZOS regression maps have been added for AMV, PDV and both IPV versions.

4) ZOS has been added to the list of variables for the climatological averages, standard deviation and trend maps.

5) A number of new regional time series have been added.

**Specific comments**

Section 3.1: It would be helpful to include some technical details about the CVDP, such as the programming language it is implemented in and its compatibility with different operating systems.

The CVDP is written in NCL (The NCAR Command Language). NCL can be installed on most commonly used operating systems. We hope to have a python version of the CVDP available within the next year.

We have added the following text '*The CVDPv6 is written in NCL (The NCAR Command Language), which can be installed on most commonly used operating systems*' on line 144

Section 3.3: While I was able to access the CVDPv6 output for MMLEAv2 at https://webext.cgd.ucar.edu/Multi-Case/MMLEA_v2/ as indicated in the section, the current hosting approach—being a directory on a web server—raises concerns about long-term stability and version control. Compared to more permanent data archiving solutions such as DOI-based repositories (e.g., Zenodo), this method may not ensure robust traceability. Are there any plans to improve the long-term accessibility and versioning of these data products?

This is a good point. CVDP output (available from the CVDP Data Repository) has been available from NCAR's website for 10+ years now, and up to this point no one has questioned whether this is the best way to go about distributing it. Currently we do not offer versioning of the output data. We will consider both versioning and increasing accessibility in the future.

It would be beneficial to discuss potential linkages and/or interaction plans with other established evaluation tools, such as the Earth System Model Evaluation Tool (ESMValTool; Eyring et al., 2016), PCMDI Metrics Package (PMP; Lee et al., 2024), and Climate Model Assessment Tool (CMAT; Fasullo, 2020). In my knowledge, CVDP contributes to ESMValTool, and similar capability is also available in the PMP that can help (or already helped?) cross validation of the tools. Addressing these (potential) connections recievedwould provide readers with a

comprehensive understanding of CVDP's role within the broader ecosystem of climate model analysis and evaluation tools.

An earlier version of the CVDP can indeed be run through ESMValTool. The CVDP can also be run from CESM's AMWG Diagnostics Framework (ADF) package and soon from within the CESM Unified Postprocessing and Diagnostics (CUPiD) package. Amongst many other uses, the CVDP is used alongside the CMAT package to evaluate CESM within the model's development cycle.

**Figures:** Some multi-panel figures appear overly complex and difficult to read due to their small size. Is there a way to simplify these figures while maintaining clarity? For instance:

- In **Figure 7**, the middle column seems to repeat the same plot (ERA5_1) across multiple rows.
- In **Figure 8**, the second column (MEM) appears to contain redundant plots.
- In **Figure 9**, the third column (MEM Change) also seems repetitive.
- **Figures 4 and 5** may similarly benefit from reorganization.

The authors could consider reducing the number of plots by eliminating redundancies and restructuring the panel layout to improve readability.

We understand the redundancy but want to show what is in the CVDP and how it plots the output so we prefer to leave these as they are.

Overall, this manuscript provides an important resource for the climate modeling community. Addressing the points above would further enhance its clarity, impact, and accessibility.

**References**

Eyring, V., Righi, M., Lauer, A., Evaldsson, M., Wenzel, S., Jones, C., Anav, A., Andrews, O., Cionni, I., Davin, E. L., Deser, C., Ehbrecht, C., Friedlingstein, P., Gleckler, P., Gottschaldt, K.-D., Hagemann, S., Juckes, M., Kindermann, S., Krasting, J., Kunert, D., Levine, R., Loew, A., Mäkelä, J., Martin, G., Mason, E., Phillips, A. S., Read, S., Rio, C., Roehrig, R., Senftleben, D., Sterl, A., van Ulft, L. H., Walton, J., Wang, S., and Williams, K. D.: ESMValTool (v1.0) – a community diagnostic and performance metrics tool for routine evaluation of Earth system models in CMIP, Geosci. Model Dev., 9, 1747–1802, https://doi.org/10.5194/gmd-9-1747-2016, 2016.

Fasullo, J. T.: Evaluating simulated climate patterns from the CMIP archives using satellite and reanalysis datasets using the Climate Model Assessment Tool (CMATv1), Geosci. Model Dev., 13, 3627–3642, https://doi.org/10.5194/gmd-13-3627-2020, 2020.

Lee, J., Gleckler, P. J., Ahn, M.-S., Ordonez, A., Ullrich, P. A., Sperber, K. R., Taylor, K. E., Planton, Y. Y., Guilyardi, E., Durack, P., Bonfils, C., Zelinka, M. D., Chao, L.-W., Dong, B., Doutriaux, C., Zhang, C., Vo, T., Boutte, J., Wehner, M. F., Pendergrass, A. G., Kim, D., Xue, Z., Wittenberg, A. T., and Krasting, J.: Systematic and objective evaluation of Earth system models: PCMDI Metrics Package (PMP) version 3, Geosci. Model Dev., 17, 3919–3948, https://doi.org/10.5194/gmd-17-3919-2024, 2024.

---

## Author Response (AR3)

The updated manuscript has been returned to me after a second round with reviewers. One reviewer has made a couple of minor comments which I post below in case you don't have access to the report. Please respond to these comments on the open discussion page of the manuscript. I look forward to receiving your revised manuscript.

Regards,

Penny

*We thank the anonymous reviewers and the editor for their time and comments, In addition to our response below we note that we have also added the following.*

*Thermocline depth for as many models as possible (see Table 1). We note that this data has taken much longer to process than planned due to the unexpected outage of our supercomputing facility. This data is now fully processed and is currently being uploaded to the public archive. Should the editor wish to wait to this upload to be complete before accepting this review – we completely understand. We have also added Jemma Jeffree as an author to acknowledge her contribution of processing 3D ocean temperatures to get this depth.*

*We have added the GISS-E2 model which is described in the text on lines 126-133.*

Line 4: Observations sitting within the ensemble spread is not a sufficient test of the models - the credibility of the model's response to forcings and simulation of internal variability is crucial. This needs to be highlighted in the abstract to prevent misinterpretation of the models.

*The abstract highlights what we did in the paper as such we have left this as is. We note that we talk about a fair comparison of models and observations on line 4 not an evaluation of the models credibility for it's simulation of individual factors. We have added additional information around this point on line 27 expanding on the information in the abstract which reads 'To this point, fairly evaluating projections in single runs of climate models, particularly for highly variable climate quantities against this single realisation of the real world is only possible by taking long time averages, to effectively smooth out natural climate variability and allowing for the assessment of the model's forced response. The advantage of using a LE is that we can additionally evaluate whether observations sit within the model's ensemble spread. This is a necessary, although not sufficient, condition for model evaluation that makes LEs invaluable tools for such evaluation.'*

Lines 36-51: Thanks for adding this text. However, the need to test for potential signal to noise errors should be explicitly highlighted, because if they exist it will fundamentally undermine the straightforward interpretation of large ensembles. Furthermore, whilst I completely agree with the aims of ForceSMIP, if I understand the protocol by using model simulations to evaluate the methods it will not assess signal to noise errors. If so, please make this clear.

*Line 52 has been added to read ' Finally, the recent discovery of the signal-to-noise paradox (Scaife and Smith, 2018) where an ensemble can better predict observations than their own members, is also an avenue where LEs can provide valuable insight into model behaviour (Weisheimer et al., 2024).'*

Line 179: please add 'modelled' in front of 'response to external forcing'

*This has been done.*

Line 270: but large ensembles are necessary to test for signal to noise errors. Please amend this

paragraph taking this into account.

Lines 369-370: but large ensembles are needed to test for signal to noise errors - please amend

*For the previous two comments we agree – large ensembles are needed to test for signal to noise errors, to quantify variability and potentially many other factors. We do not argue that this is not the case. We argue that for a fair comparison of a observational value to a modelled one if the variability is low, we need less ensemble members. We have changed the wording to comparison in both cases rather than assessment/evaluation to make this point clearer.*

Lines 373-376: but detrending does not remove higher frequency signals forced by aerosols, volcanoes, solar, ozone etc. Please be clear about this.

*Dentrending by removing a quadratic fit does not remove these signals, removing the ensemble mean does. This is why we highlight that both methods can be used in the package and compared with each other.*

Fig 4: I needed 400% magnification to read the text - would it be possible to make it clearer please?

*This will be rectified when the Figure is bigger in the final manuscript and the caption can be moved to make more room for the Figure.*

Fig 6: the caption still says 'The dark blue curve shows the model's ensemble mean timeseries' but surely it is the ensemble mean power spectrum?

*Thanks for catching this – we have fixed it.*

---

## Author Response (AR4)

Dear Editor,

Please note that on a final proofread we found a few typos which were fixed in this version compared to the accepted one.

I confirm that nothing else was changed in this new version.

Nicola Maher